# Wearable device-based health equivalence of different physical activity intensities against mortality, cardiometabolic disease, and cancer

Raaj Kishore Biswas [1,2,5], Matthew N. Ahmadi [1,2,5], Adrian Bauman[3], Karen Milton[4], Nicholas A. Koemel [1,2] & Emmanuel Stamatakis [1,2] ✉

Current conventions, partly derived from self-reported data, typically equate 1 minute of vigorous physical activity (VPA) to 2 minutes of moderate physical activity (MPA). Using accelerometer-derived intensity classification in 73,485 UK Biobank participants (mean follow-up: 8.0 [1.0] years), we assess the equivalence of light activity (LPA) and MPA to 1 minute of VPA for all-cause (ACM) and cardiovascular (CVD) mortality, major adverse cardiovascular events (MACE), type 2 diabetes, and cancer outcomes. For a standardised 5%–35% risk reduction, the median MPA equivalent per minute of VPA is 4.1 (ACM, 95% CI: 4.1–4.2), 7.8 (CVD mortality, 7.7-8.0), 5.4 (MACE, 5.3–5.5), 9.4 (type 2 diabetes, 9.3-9.6), and 3.5 (cancer mortality, 3.4-3.5) minutes. For non-cancer outcomes, the median LPA equivalent per 1 minute of VPA ranges from 53 (ACM) to 94 minutes (type 2 diabetes), reflecting generally weaker dose-response curves of LPA with all outcomes. These findings indicate a substantial departure from self-reported estimates and support integrating device-based equivalence into guidelines and wearables.

The benefits of physical activity are well established for both the general population and for specific clinical subgroups[1,2]. There is a clear dose-response relationship between physical activity intensity and mortality or non-communicable disease risk reduction[1,3,4]. Based on such evidence, the current WHO physical activity guidelines recommend that all adults accumulate 150–300 min of moderate-intensity physical activity (MPA) or 75–150 min of vigorous physical activity (VPA) per week or an equivalent combination of moderate and vigorous activities[1]. This recommendation is primarily based on epidemiological evidence from self-reports, such as questionnaires[5,6], where each minute of vigorous activity is given twice the weight of moderate-intensity activities, representing a 1:2 ratio. This convention is based on metabolic equivalent (MET) values whereby each vigorous activity (>6 MET)[7] requires, on average, about twice the energy expenditure of moderate activities (3−6 MET)[7]. Studies examining the associations of physical activity volume with mortality and other long term health outcomes[8,9] have thus assumed that 1 min of vigorous activity produces similar benefits to around 2 min of moderate activity. However, this is non-empirically derived and has not been objectively tested on a population level despite the emergence of highly granular wearables data in health research. Additionally, no studies have compared light intensity with moderate or vigorous activity to determine their equivalence.

[1]Mackenzie Wearables Research Hub, Charles Perkins Centre, The University of Sydney, Sydney, NSW, Australia. [2]School of Health Sciences, Faculty of Medicine and Health, The University of Sydney, Sydney, NSW, Australia. [3]School of Public Health, Faculty of Medicine and Health, The University of Sydney, Sydney, NSW, Australia. [4]Norwich Medical School, University of East Anglia, Norwich, UK. [5]These authors jointly supervised this work: Raaj Kishore Biswas, Matthew N. Ahmadi. ✉e-mail: emmanuel.stamatakis@sydney.edu.au

Recent device measured studies have demonstrated that vigorous activity may be considerably more time efficient than moderate or light intensity activities in producing cardiovascular and metabolic adaptations[10,11], a finding that is also reflected by epidemiological studies examining the associations of vigorous activity with CVD and mortality[4,12,13]. These studies indicate that there may be lower time commitment and convenience benefits for structured and intermittent vigorous activities for both patients and healthy populations, although vigorous exertion may cause discomfort and pose adherence challenges in people who are not accustomed to it[14]. To expand intervention options, further work is needed to specifically quantify the "health value" of activities of different intensity levels against key health outcomes.

Fewer studies have examined if light-intensity physical activity (LPA) is associated with prospective health outcomes[15], in part due to the inability of self-reported instruments to capture light intensity activity[16,17]. Systematic reviews indicate some potential for reducing all-cause mortality risk from light intensity activity, and randomized controlled trials show improvements in glycemic control and other metabolic parameters[18,19]. The few studies that used devices to quantify the association of LPA with mortality or cardiovascular disease show mixed findings, and none estimated specifically its per-time unit equivalence to vigorous intensity[15,20,21].

Using algorithms that are partly based on existing, less empirical, research standards[3,8,9], consumer grade wearables commonly assign a "health value" to each minute of physical activity the wearable device records, to support setting and monitoring physical activity goals[22,23], e.g., Google Fit Heart Points and Apple Fitness+[22,23]. For example, Google Fit assigns Heart Points based on the intensity of the activity: 1 heart point if it is moderate intensity (100–130 steps per minute) and 2 heart points if it is vigorous (>130 steps per minute)[24]. In the absence of published epidemiological studies estimating the equivalence of wearable device-measured physical activity intensity, it is highly unlikely that any consumer-grade algorithms reflect the associations of intensity with long term clinical and mortality endpoints when calculating these metrics.

In this work, we used a large device-based population cohort and a validated two-stage machine learning-based intensity classification schema to examine the equivalence of light and moderate intensity against each minute of vigorous intensity against a range of key mortality, cardiometabolic, and cancer related outcomes. Each minute of vigorous intensity activity is roughly equivalent to 4–9 min of moderate and 53–156 min of light intensity physical activity, suggesting a considerable departure from previous self-reported evidence conventions. These findings support the incorporation of device-based health equivalence estimates into future physical activity guidelines and the algorithms used in consumer-grade wearables.

## Results
### Cohort information
The UK Biobank is a prospective cohort study involving adults aged 40–69 at baseline (2006–10). At baseline, the mean age of participants was 61.6 years (SD 7.9), 41,420 (56.4%) were women and 32,065 (43.6%) were men.

From 2013 to 2015, a total of 103,684 UK Biobank participants wore a wrist-worn accelerometer on their dominant wrist for 24 h a day over a period of 7 days[25], in line with previous studies[12,13]. Our analysis included participants who had a minimum of 3 valid wear days (≥16 h of wear-time per day), with at least one of those days being a weekend day. We excluded those with insufficient valid wear days, missing covariate data, or who reported an inability to walk. Supplementary Fig. 1 provides a flow diagram of the study participants. We followed the Strengthening the Reporting of Observational Studies in Epidemiology (STROBE) reporting guideline (Supplementary Data 4).

Our analytic sample included 73,485 participants with 2675 all-cause mortality events, 545 CVD mortality events, 2359 MACE events, 1836 type-2 diabetes events, 538 physical activity related cancer mortality events, and 2662 physical activity related cancer incidence events. Supplementary Data 1 shows participant characteristics by levels of VPA duration. The mean age of participants was 61.6 (7.9) years and mean follow-up was 8.0 (1.0) years, corresponding to 585,313 (ACM), 538,708 (CVD mortality), 531,656 (MACE), 564,048 (type 2 diabetes), 551,512 (physical activity related cancer mortality), and 541,893 (physical activity related cancer incidence) person-years.

### Dose-response associations of intensity-specific physical activity
Supplementary Fig. 2 illustrates the adjusted dose-response relationships for VPA, MPA, and LPA after mutual and multivariable adjustments. Compared to the referent (VPA and MPA: 0 min/day; LPA: 37.9 min/day), both VPA and MPA demonstrated favourable dose-response associations across all outcomes, whereas LPA exhibited a slight risk reduction for type 2 diabetes and ACM only. Specifically, VPA displayed pronounced dose-response gradients for all outcomes, exhibiting nearly linear associations for ACM, CVD mortality, MACE, and type 2 diabetes, with hazard ratios reaching 50% or higher. MPA showed a steep inverse dose-response relationship with all outcomes up to around 30 min per day. In contrast, LPA demonstrated a subtle gradient with type 2 diabetes and ACM, with no statistically significant associations observed for any other outcomes.

### Equivalence of different physical activity intensities
Supplementary Data 2 and 3 presents the risk reduction for all six outcomes, in 5% increments for each physical activity intensity. Figures 1 and 2 describe the equivalent minutes of MPA and LPA relative to each minute of VPA for risk reduction within the range of 5–35% (hazard ratio: 0.95–0.65) through continuous increments.

We observed a wide range of equivalence between MPA and VPA across outcomes. The median equivalence of MPA per minute of VPA was 4.1 (95% CI: 4.1–4.2) minutes for ACM, 7.8 (7.7–8.0) minutes for CVD mortality, 5.4 (5.3–5.5) minutes for MACE, and 9.3 (9.3–9.6) minutes type 2 diabetes. The VPA:MPA equivalence curve (Fig. 1) remained stable for ACM, MACE, and CVD mortality, while the curve for type 2 diabetes shifted upwards for higher levels of risk reduction (>25%). The median MPA equivalence for each minute of VPA was 6.6 min for ACM, CVD mortality, MACE, and type 2 diabetes. For physical activity related cancer mortality and incidence, the dose-response association was weaker than other outcomes (Supplementary Fig. 2), as a result the equivalence between VPA and MPA was lower, with ratios of 1 to 3.5 for physical activity related cancer mortality and 1 to 1.6 for physical activity related cancer incidence.

In contrast to MPA, the LPA to VPA equivalence exhibited considerable variability across different levels of risk reduction (Supplementary Data 2–3 and Fig. 2). For non-cancer outcomes the median equivalence of LPA per minute of VPA ranged from 53 (ACM) to 92 minutes (type 2 diabetes). For ACM we observed an equivalence of 52.6 (51.9-53.5) minutes of LPA for every 1 min of VPA, 72.5 (71.7–73.7) minutes for CVD mortality, 86.1 (84.4-87.9) minutes for MACE and 94.0 (92.3–95.3) minutes for type 2 diabetes. The median LPA equivalence for each minute of VPA was 79.3 min for ACM, CVD mortality, MACE, and type 2 diabetes. Due to the weak dose-response association of LPA with physical activity-related cancer outcomes, the equivalence with VPA was also markedly different to other outcomes (e.g., 1 min of vigorous to a median of 156.2 (153.3–159.3) minutes of LPA for physical activity related cancer mortality, Supplementary Data 3 and Fig. 2). Using the same dose response data, we further derived the per MPA minute equivalence of LPA in terms of MPA by dividing the VPA-to-LPA and VPA-to-MPA ratios (Table 1). In summary, the median LPA equivalence per 1 min of MPA ranged from 3.1 min (physical activity

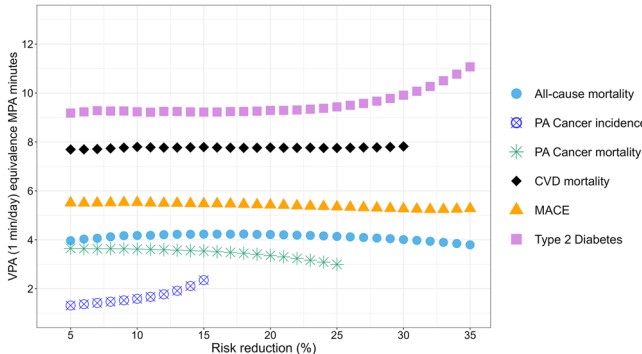

**Fig. 1 | Equivalence of moderate (MPA) intensity time against 1 min of vigorous intensity physical activity (VPA) by increments of risk reduction in all-cause mortality, major adverse cardiovascular events (MACE), CVD mortality, type 2 diabetes, physical activity related cancer mortality and physical activity related cancer incidence.** All-cause mortality: $n = 73,485$; events = 2675; CVD mortality: $n = 67,499$; events = 545; MACE: $n = 67,314$; events = 2359; Type 2 diabetes: $n = 71,533$; events = 1836; physical activity related cancer mortality: $n = 69,115$; events = 538; physical activity related cancer incidence: $n = 68,828$; events = 2662. Risk reduction is based on the hazard ratios (HR) of the dose-response curves for each outcome. Analyses were adjusted for sex, age, education, ethnicity, fruit and vegetable consumption, smoking history, alcohol consumption, sleep duration, discretionary screentime, cardiovascular disease (CVD) related medication use (insulin, blood pressure, cholesterol) and family history of cancer and CVD. For ACM and type 2 diabetes, analyses were adjusted for a previous cancer and CVD diagnosis; for MACE and CVD, analyses were adjusted for previous cancer incidence; and for cancer, analyses were adjusted for previous CVD diagnosis. Each physical activity intensity-specific spline model was mutually adjusted for physical activity energy expenditure from other intensities. Referent data point was set to zero for VPA and MPA splines. All analyses excluded participants who had an event in the first year of follow-up and prevalent major CVD diagnosis at or prior to the accelerometry baseline for MACE and CVD mortality outcomes. Cancer analyses similarly excluded diagnosed cases of cancer at or prior to the accelerometry baseline. Type 2 diabetes analyses excluded prevalent cases at or prior to the accelerometry baseline.

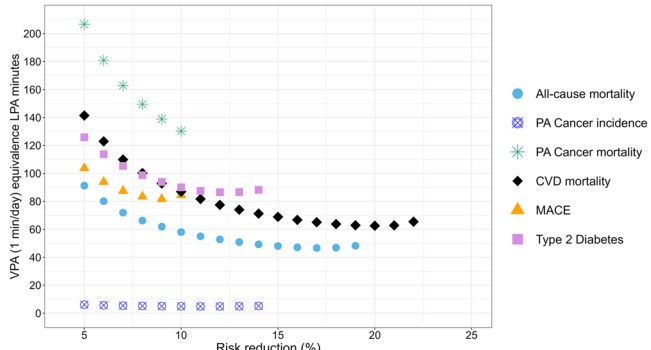

**Fig. 2 | Equivalence of light (LPA) intensity time against 1 min of vigorous intensity physical activity (VPA) by increments of risk reduction in all-cause mortality, major adverse cardiovascular events (MACE), CVD mortality, type 2 diabetes, physical activity related cancer mortality and physical activity related cancer incidence.** All-cause mortality: $n = 73,485$; events = 2675; CVD mortality: $n = 67,499$; events = 545; MACE: $n = 67,314$; events = 2359; Type 2 diabetes: $n = 71,533$; events = 1836; physical activity related cancer mortality: $n = 69,115$; events = 538; physical activity related cancer incidence: $n = 68,828$; events = 2662. Risk reduction is based on the hazard ratios (HR) of the dose-response curves for each outcome. Analyses were adjusted for sex, age, education, ethnicity, fruit and vegetable consumption, smoking history, alcohol consumption, sleep duration, discretionary screentime, cardiovascular disease (CVD) related medication use (insulin, blood pressure, cholesterol) and family history of cancer and CVD. For ACM and type 2 diabetes, analyses were adjusted for a previous cancer and CVD diagnosis; for MACE and CVD, analyses were adjusted for previous cancer incidence; and for cancer, analyses were adjusted for previous CVD diagnosis. Each physical activity intensity-specific spline model was mutually adjusted for physical activity energy expenditure from other intensities. Referent data point was set to zero for VPA splines and the minimum value of LPA (37.9 min) for LPA splines. All analyses excluded participants who had an event in the first year of follow-up and prevalent major CVD diagnosis at or prior to the accelerometry baseline for MACE and CVD mortality outcomes. Cancer analyses similarly excluded diagnosed cases of cancer at or prior to the accelerometry baseline. For type 2 diabetes previous diagnosed cases at or prior to the accelerometry baseline were excluded.

related cancer incidence), to 1:9.3 min (CVD mortality), 10.0 min (type 2 diabetes), 12.9 min (ACM), 15.8 min (MACE), and 45.0 min (physical activity related cancer mortality).

## Discussion

This large-scale study of 73,485 adults, aged 40 to 79 years at accelerometry baseline, provides, to our knowledge, the first examination of physical activity intensity equivalence with a broad range of major outcomes including all-cause, cardiovascular, and cancer mortality. We found 1-min of vigorous intensity physical activity was equivalent to about 4–9 min of moderate intensity and 53–156 min of light intensity for all-cause mortality and cardiometabolic outcomes (MACE, type 2 diabetes, and CVD mortality). Our findings are in stark contrast to the widely used current convention of a 1:2 ratio between vigorous and moderate intensity, which was derived from self-reported data. Using a wide array of mortality and major non-communicable disease end points, our study is the first to systematically quantify the equivalence between LPA, MPA, and VPA various intensities of physical activity for use in public health and prevention research, as well as in consumer-grade wearable devices. Our work expands behaviour change options by indicating alternatives to higher intensities, informs the development of physical activity guidelines based on wearable devices, enhances surveillance measures, and supports practitioners and future trials to establish more accurate dosages and define options for prescribing physical activity prescription and personalised medicine initiatives.

For all-cause mortality, we observed a consistent equivalence ratio of 1:4 for vigorous intensity to moderate intensity when the

inverse dose-response association ranged between 5% to 35% lower mortality. For cause-specific mortality of CVD and cancer, the equivalence ratio was slightly stronger, where we observed a consistent ratio of 1:8 and 1:3.5 for vigorous intensity to moderate intensity. These findings suggest a 2–4 times stronger potential health effect for vigorous intensity minutes than previously observed in cohort studies reliant on self-report[26–28]. The discrepancy in our findings compared to traditional self-report evidence is likely due to the higher precision afforded by the continuous and highly granular (10 s epochs) physical activity estimation provided by our wearable data methodology[4,12,13], which enabled us to quantify the time spent in different intensity ranges. Self-report methods rely on participants recalling blocks of time spent in moderate to vigorous activities (usually structured exercise) that last above a certain minimum duration threshold, typically 10 to 15 minutes and over[29]. This can lead to various biases, such as regression dilution bias[30], and reflects the inherent imprecision of such measurements. The previously recommended 1:2 ratio between vigorous and moderate intensity is likely the result of such limitations of self-reports[10].

Our equivalence findings between vigorous and moderate intensity are broadly consistent with randomised controlled trials and cross-sectional studies. Previous works that have compared vigorous intensity to moderate intensity activities and exercise under supervised and controlled clinic conditions[31–35] demonstrated an approximate 1:7 to 1:13 ratio for surrogate cardiometabolic outcomes such as lipid profile, blood pressure, and cardiorespiratory fitness [31,36–39]. Our results expand this evidence with long-term clinical endpoints and mortality in a real-world population cohort measured in environments outside

**Table 1 | Summary of median equivalence against the study outcomes across all three intensity bands (N = 73,485)**

|  | VPA: MPA: LPA[a] | VPA: MPA[a] | VPA: LPA[a] | MPA: LPA[b] |
|---|---|---|---|---|
| Outcomes |  |  |  |  |
| All-cause mortality | 1: 4.09: 52.65 | 1: 4.09 | 1: 52.65 | 1: 12.87 |
| CVD mortality | 1: 7.78: 72.47 | 1: 7.78 | 1: 72.47 | 1: 9.31 |
| MACE | 1: 5.44: 86.13 | 1: 5.44 | 1: 86.13 | 1: 15.83 |
| Type 2 diabetes | 1: 9.40: 94.03 | 1: 9.40 | 1: 94.03 | 1: 10.00 |
| Physical activity related cancer mortality | 1: 3.47: 156.23 | 1: 3.47 | 1: 156.23 | 1: 45.02 |
| Physical activity related cancer incidence | 1: 1.63: 5.09 | 1: 1.63 | 1: 5.09 | 1: 3.12 |

*VPA* Vigorous Physical Activity, *MPA* Moderate Physical Activity, *LPA* Light Physical Activity, *CVD* Cardiovascular disease, *MACE* Major adverse cardiovascular events.
[a]Calculated using dose-response curves presented in Supplementary Fig. 2 and median equivalence estimate reported in Supplementary Data 2-3.
[b]Calculated by dividing the estimated VPA-to-LPA and VPA-to-MPA ratios.

controlled clinic conditions. Our findings at the population level alongside those of smaller RCTs under controlled conditions provides evidence that may improve future public health messaging and programs. Taken together, our findings prompt some reinterpretation of the physical activity evidence-base, particularly in regard to current guidelines suggesting 75–150 min of vigorous intensity is equivalent to 150–300 min of moderate intensity, reflecting the traditional 1:2 ratio between vigorous and moderate intensity.

We observed an equivalence ratio of 1:53 for vigorous intensity to light intensity for all-cause mortality and a ratio of 1:73 for cardiovascular and 1:156 for cancer mortality. Due to the inability of questionnaires to capture light intensity activities, device-based assessment provides the first direct examination of light intensity time equivalence. Although light intensity is accumulated in higher volumes throughout a standard day, our findings highlight the time efficiency of vigorous activity, with requiring 1–2 h of light intensity activity to derive analogous associations as 1 min of vigorous intensity. In addition, for all outcomes, light intensity did not lower risk to an equivalent dose-response association beyond 15% for ACM, MACE, and type 2 diabetes, and 10% for physical activity related cancer. These findings are particularly relevant for adults at high risk of disease, or those who are time poor, and suggest different health promotion strategies. For example, exercising at different intensities or completing light intensity daily activities such as household tasks, stair climbing, or light strolling could be interspersed with short bursts[13,32] of fast walking for a more impactful physical activity session. Our findings provide clinicians with essential information for selecting intervention options to support physical activity behaviour change in patients or individuals at high risk of major lifestyle-related chronic diseases. Specifically, consumer wearable devices that provide activity feedback may facilitate behaviour change, and understanding how accumulation across different activity intensities can be used to tailor treatment.

The prolonged times spent in light intensity activity in this cohort, and the weak dose-response association we observed, are relatively consistent with prior device-based studies and meta-analyses that have examined light intensity[15,20,33,34]. Prior examinations of bias effects in population cohorts have found light intensity activity is more prone to reverse causation than higher intensity activities[35]. The precautions we took to mitigate the influence of such bias in our analyses may explain, in part, the observed weaker dose-response associations of light intensity compared to other studies[40,41] that found a protective association[42,43].

The equivalence estimates are derived from predicted hazard ratios (HRs) across different intensity bands, using a single Fine-Gray subdistribution hazard model for each disease outcome. To quantify uncertainty around these estimates, we applied a bootstrapping approach to the predicted values. This generated confidence intervals that represent the variability inherent in the model-based predictions. It is important to note that these confidence intervals are not intended to serve as indicators of statistical significance. Rather, they are

presented to illustrate the range of plausible equivalence estimates based on the model's assumptions and sampling variability. They should not be interpreted as indicators of statistical significance.

Strengths of our study include the use of wearables to quantify physical activity at a high granularity in the largest resource to date with linkage to prospective health outcomes. We examined a spectrum of conditions responsible for much of the burden of non-communicable disease, encompassing both incidence and mortality, to explore equivalence across specific categories of physical activity intensity. The large sample size and long follow up allowed us to reduce the risk of reverse causality by removing participants who had an event within the initial 12 months of follow-up or prevalent disease at baseline. Despite the extensive precautionary measures, the potential for our equivalence estimates to reflect a degree of reverse causation may still exist, for example low activity levels due to undiagnosed or prodromal disease[35,44]. Due to the observational design, we cannot rule out the presence of unmeasured confounding. However, our adjustment for potential confounders was based on previously established cause-effect pathways between physical activity and non-communicable disease that informed our direct acyclic graph. The wearables used in the UK Biobank relied on accelerometers quantifying absolute intensity, not energy expenditure. Previous research, however, has indicated that adjusting absolute intensity for participant characteristics that determine energy expenditure (weight, height, age, and sex), does not materially influence the association of physical activity volume with cause-specific and overall mortality[40]. The median time gap between the UK Biobank baseline, when covariate measurements were collected, and the accelerometry study was 5.5 years. However, most covariates remained stable over this period, with the exception of medication use[13]. The UK Biobank had a very low response rate (5.5%), and participants in our sample were subject to additional selection criteria which should be considered when interpreting our results. However, evidence suggests that, at least with mortality endpoints, low response rates, and the subsequent unrepresentativeness to the target population, does not materially affect estimates of the association of physical activity[41].

Our study represents the first device-based evidence on the health equivalence of different physical activity intensities. Our findings reveal a distinctively higher equivalence for moderate intensity compared to the previous convention based on self-reported data, which equates one minute of vigorous intensity to 2 min of moderate intensity, as reflected in current guidelines[1,45,46]. Each minute of vigorous activity is associated with risk reduction levels comparable to 4–9 min of moderate intensity; and 53–156 min of light intensity physical activity - while acknowledging that not even the largest amounts of daily LPA can elicit the beneficial effects of moderate or vigorous intensities. Our findings inform future guidelines and lifestyle interventions, and can help improve the algorithms used in consumer-grade wearables to quantify the health benefits of each minute of physical activity or exercise.

## Methods

### Physical activity assessment and exposure variables

The data were calibrated, and non-wear periods were identified according to standard procedures[47,48]. Non-wear periods were identified using the signal standard deviation to detect periods where the standard deviation was below 0.13 milligravity for 30 or more minutes, reflecting non-wear. Sleep was detected based on relative changes in wrist tilt angle.

We categorised physical activity intensities into LPA, MPA and VPA using a validated two-stage machine learning-based Random Forest activity classifier[4,13]. We have described the physical activity intensity classification schema in detail elsewhere[12,13,32] and we include here as Supplementary text (including Supplementary Figs. 3–4 and Supplementary Tables 7–8). In summary, this 2-stage machine learning-based Random Forest activity classifier uses raw acceleration signals to identify and quantify time spent in different activity types and intensities in 10 s windows.

### Mortality and disease endpoints

Definitions of outcomes along with disease specific International Classification of Diseases codes are provided in Supplementary Table 5. In addition to all-cause mortality (ACM), five other outcomes were derived: CVD mortality, major adverse cardiovascular events (MACE), type 2 diabetes, physical activity related cancer mortality and incidence. CVD was defined as diseases of the circulatory system, excluding hypertension, diseases of arteries, and lymph. MACE was defined as CVD death or incidents of myocardial infarction, stroke, and heart failure. Incidents of type 2 diabetes was extracted from hospitalisation and general practitioner records. Physical activity related cancer encompasses various cancer types associated with increased levels of physical activity, excluding in situ, benign, uncertain, non-melanoma skin cancer, or non-well-defined cancers[49]. The selected cancer types were informed by prior meta-analyses and pooled studies that demonstrated significant or suggestive associations between physical activity and cancer risk, see detailed list of cancer ICD-10 codes in Supplementary Table 5.

### Events ascertainment

Participants were followed up through November 30th, 2022, with deaths obtained via linkage with the National Health Service (NHS) Digital of England and Wales or the NHS Central Register and National Records of Scotland. Inpatient hospitalisation data were provided by either the Hospital Episode Statistics for England, the Patient Episode Database for Wales, or the Scottish Morbidity Record for Scotland. Cancer data linkage was obtained through national cancer registries. For England and Wales, cancer diagnosis data were followed up through 31 December 2020 and 31 December 2016 respectively, and were provided by NHS England[50]. For Scotland, cancer diagnosis data were followed up through 30 November 2021 and provided by the National Records of Scotland[50].

### Inclusion criteria

We followed established inclusion criteria from analogous analyses[4,12,13,32]. For each outcome, participants with a prior diagnosis of the respective condition by the time the accelerometry baseline started were excluded. Adjustments in the models primarily accounted for cardiovascular disease (CVD) and cancer, as these are the leading causes of mortality. Specifically, analyses for all-cause mortality (ACM) and type 2 diabetes were adjusted for prior diagnoses of cancer and CVD. For major adverse cardiovascular events (MACE) and other CVD outcomes, prior cancer incidence was considered, while analyses for cancer outcomes were adjusted for previous CVD diagnoses.

### Statistics and reproducibility

To reduce the risk of reverse causation due to undiagnosed disease, individuals registering an event within the initial 12 months of follow-up were excluded from diseases specific analyses[4,32]. We also excluded those with prevalent diseases at baseline accelerometry assessment[4,13,32]. The upper range of all three physical activity intensity-specific categories was winsorized at the 97.5th percentile to alleviate the impact of sparse data or outliers[4,13,32].

We investigated the time-to-event dose-response associations of intensity-specific categories with the six outcomes. For these analyses, we calculated hazard ratios (HRs) using Cox proportional hazards for ACM, and Fine-Gray sub-distribution hazard models for all disease-specific outcomes to account for competing risks from deaths not attributed to the analytic outcome[51]. Knots were placed at the 10th, 50th and 90th percentiles[52]. Departure from linearity was assessed by a Wald test. Proportional hazards assumptions were tested using Schoenfeld Residuals in the models and no violations were observed (all $p > 0.05$).

In line with previous analogous analyses[4,12,13,32], core analyses were adjusted for sex, age, education, ethnicity, fruit and vegetable consumption, smoking history, alcohol consumption, sleep duration, discretionary screentime, CVD related medication use (insulin, blood pressure, cholesterol) and family history of cancer and CVD. Each physical activity intensity-specific spline model was mutually adjusted for physical activity energy expenditure-based volume from other intensities, for example the vigorous activity splines were adjusted for energy expenditure-based volume of moderate and light intensity activity. Referent data point was set to the minimum for all three intensity-specific splines (0 min for VPA and MPA, and 37.9 min for LPA)[20,53]. Complete covariate definitions are provided in Supplementary Data 6.

For estimating equivalence across the three physical activity intensity-specific categories, we extracted the hazard ratios from the Cox/Fine-Gray sub-distribution models. We compared the intensity values within the 5–35% risk reduction range based on the observed relative uniformity in the shape of the dose response curves. An additional reason this range was chosen was that it represents a meaningful spectrum of risk reduction observed across most intensity bands and the selected outcomes. For example, for any given percentage risk reduction in ACM, the hazards of VPA were compared with the same percentage reduction of MPA and LPA. Subsequently, the risk-specific MPA/LPA values (in minutes) were standardized by dividing them by the corresponding VPA values, thereby equating MPA and LPA to one minute of VPA. We calculated 95% confidence intervals through non-parametric bootstrapping of the model-predicted values using 1000 resamples. For each resample, the outcome variable was estimated at rounded predicted risk levels (e.g., 0.65, 0.70,..., 0.95), and the 2.5th and 97.5th percentiles of the resulting bootstrap distributions were used to derive the confidence bounds[4,54].

We conducted all analyses using R statistical software (*version 4.2.3*), employing the RMS (*version 6.3.0*) and survival packages (*version 3.5.5*). Our reporting adheres to the Strengthening the Reporting of Observational Studies in Epidemiology (STROBE) guidelines (Supplementary Data 4).

### Ethics approval

Participants gave informed consent, and ethical approval was granted by the National Health Service's National Research Ethics Service in the UK (Ref: 11/NW/0382).

### Reporting summary

Further information on research design is available in the Nature Portfolio Reporting Summary linked to this article.

## Data availability

The data that support the findings of this study are available from the UK Biobank, but restrictions apply to the availability of these data, which were used under license for the current study, and so are not publicly available. The UK Biobank data that support the findings of this study can be accessed by researchers on application (https://www.ukbiobank.ac.uk/register-apply/). Source data are provided with this paper.

## Code availability

The statistical code used in the analyses of this manuscript is available upon reasonable request.

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

## Acknowledgements

This research has been conducted using the UK Biobank Resource under Application Number 25813. The authors would like to thank all the participants and professionals contributing to the UK Biobank. This research used data assets made available by National Safe Haven as part of the Data and Connectivity National Core Study, led by Health Data Research UK in partnership with the Office for National Statistics and funded by UK Research and Innovation. This work uses data provided by patients and collected by the NHS as part of their care and support. This study is funded by an Australian National Health and Medical Research Council (NHMRC) Investigator Grant (APP 1194510). The funder had no specific role in any of the following study aspects: the design and conduct of the study; collection, management, analysis, and interpretation of the data; preparation, review, or approval of the manuscript; and decision to submit the manuscript for publication.

## Author contributions

R.K.B and M.N.A are joint first authors. E.S. generated the idea and supervised the project. R.K.B. performed the formal analysis and generated tables and figures. M.N.A. processed the accelerometry data and supervised the statistical analyses. R.K.B. and M.N.A. wrote the first draft. A.B., K.M., N.K., and E.S. participated in the discussion of results and revision of the manuscript. R.K.B. and E.S. had full access to all the data in the study and takes responsibility for the integrity of the data and the accuracy of the data analysis.

## Competing interests

E.S. is a paid consultant and holds equity in Complement 1, a US-based company whose services relate to physical activity. All other authors disclose no conflict of interest for this work.
