## [Transparent Peer Review file · Nature Communications]

Wearable Device-Based Health Equivalence of different Physical Activity Intensities against mortality, cardiometabolic disease, and cancer

Corresponding Author: Professor Emmanuel Stamatakis

Version 0:

Reviewer comments:

Reviewer #1

(Remarks to the Author)

Key results

This a large retrospective cohort study of 73,485 adults, aged 40 to 79 years investigating the equivalence of wrist-worn wearable-device derived physical activity intensity levels on 6 outcomes of all-cause (ACM) and cardiovascular disease (CVD) mortality; major adverse cardiovascular events (MACE); type 2 diabetes incidence; and cancer mortality and incidence. VPA had a significant dose-response with ACM, CVD mortality, MACE and T2D. MPA had a dose response relationship with outcomes up to 30 minutes, and LPA was significantly associated with T2D and ACM. One-minute of vigorous intensity physical activity was equivalent to about 4-9 minutes of MPA and 53-156 minutes of LPA for all-cause mortality and MACE, type 2 diabetes and CVD mortality. The methods, analyses and results are sound and provide a contribution to the field. However, I have some concerns with the implications of the findings in the Discussion. These findings are of interest on risk mitigation specific to this sample of individuals, but it does not seem these findings can support a large worldwide public health initiative, as suggested, to change PA guidelines in light of these findings.

Abstract:

The introduction discusses assessing equivalence of LPA and MPA against VPA, but doesn't discuss equivalence of these metrics on what outcomes. Please add this in.

The conclusion states that findings support wearable device data-based guidelines, which feels like a bit of a jump from the results of the study.

Introduction:

Line 55 Risk reduction for what?

Metabolic equivalent needs to have an acronym denoted in the first mention

The first paragraph needs to have more mention of the benefits of PA and intensity levels on what particular condition/disease etc. It is too broad to just state "benefits"

The lower time benefit of VPA is a noted benefit, however, authors should discuss the higher intensity levels needed to attain VPA, particularly for the "patient" population

Line 81-83 should mention equivalence of LPA and VPA on what conditions

Line 84 – What are the health values that are assigned? These should be spelled out

Line 86 – 89 The algorithms in wearables are not calculating PA metrics base don health outcomes data; This doesn't fully make sense the way it is written. Perhaps if the above comment is elaborated upon, it may make more sense the way it is currently written.

Methods:

Non-wear time procedures should be written out instead of just referenced

Can the authors list out what cancers are associated with increased PA? And also discuss why these cancers were chosen in lieu of others?

Analyses

The rationale for excluding patients with prevalent diasese (what diseases were these?) should be discussed. These exclusion criteria may be better suited in a section listing out the patient inclusion/exclusion criteria rather than in the

analyses section

Discussion:

Major concern

Given the vast majority of adults do not meet the current recommended PA guidelines, it seems counterintuitive to increase the required minutes of MPA, for example, based on these findings. Authors may give some consideration to how these findings could be practically used to support PA promotion, rather than the notion that more is better or a higher intensity level is better. The Discussion could elaborate more on this aspect, and try to better align these findings with public health messaging, or find a way to integrate them into more of a precision medicine approach.

Lines 274-276 It may be a stretch to say this is providing clinicians with information to select interventions, when interventions are not commonly broken down into LPA, MPA, VPA etc. The information given about a possible intervention in lines 271-274 is more helpful, and further examples of who the high-risk patients are and how precision medicine using this approach and these data can be used to mitigate risk for these patients.

Limitations

Authors discuss the low response rate and unrepresentativeness to the target population as not affecting the estimate of PA with mortality, but what about the other conditions? And how do these factors weigh into the development of recommendations for wearable devices and adjustments to the existing PA guidelines?

Reviewer #2

(Remarks to the Author)

Thank you for the opportunity to review this interesting theoretical study. I did appreciate the authors' attempt to consider a long-standing issue in physical activity epidemiology. However, due to some major methodological limitations, I feel this work has limited public health policy or practice relevance.

Major concern 1: the authors correctly assert that the doubling of time allocated for vigorous-intensity physical activity (vs moderate) is a rough estimate at best. However, using data collected by devices does not allow examination of differences in energy expenditure and their relative contribution to chronic disease risk and mortality. Accelerometers measure acceleration (movement in milligravities), which is an objective measure of movement. However, without other measures to give an indication of energy being expended (even a proxy like heart rate monitoring) it is impossible to estimate energy expenditure. A 40 year old male with good cardiorespiratory fitness might experience movement captured within the algorithm for 'moderate' physical activity as 'light' intensity (based on energy expended). Whereas an obese 60 year old male recording the same acceleration (which the algorithm defines as 'moderate') might actually be expending significantly higher energy. The authors may address some of this confounding through the adjustment of their models, but BMI or another measure of body composition was not included. I understand that BMI can be considered a mediator between physical activity and chronic disease, but we are still left with a very rough approximation of energy expenditure using the device-based measures. Being able to report acceleration (milligravities) comparisons for different disease endpoints is not amenable to informing public health practice. Sometimes very rough – but easily understood – concepts are more useful for health promotion.

Major concern 2: by excluding participants with prevalent chronic diseases of interest (e.g. physical activity related cancers) from the study population, the authors have attempted to answer a causal question about (mortality) outcomes that are defined conditional on post-exposure events. When estimating effects of physical activity intensities on cancer-related mortality, participants are only eligible for inclusion if they survive to the development of cancer (which must occur before cancer-related death). Depending on when follow-up commences this can introduce selection bias and/or immortal time bias. These biases can change effect estimates considerably.

Minor comment: the authors explain in the methods that they excluded participants with prevalent chronic diseases of interest. However, some models adjust for a cancer diagnosis, or diabetes. Does this mean that the adjustment is based on incident disease that occurred after the accelerometer baseline assessment? Please clarify this point.

Version 1:

Reviewer comments:

Reviewer #1

(Remarks to the Author)

Thank you for addressing all the requested edits.

Reviewer #2

(Remarks to the Author)

I think the authors for the time they have taken to respond to my comments.

However, I think they have misunderstood my first point: even though accelerometry is an improvement upon self-reported data in many ways, the assignment to categories of 'light', 'moderate' and 'vigorous' is still based on assumptions that may not be correct at the individual level. But without collapsing the data in such a way accelerometer data are meaningless to consumers I simply wanted the authors to explain more thoroughly why the categories of accelerometer-measured activity, and the relative volumes compared to vigorous activity, will improve public health policy and programmes.

The guidelines based on self-reported estimates are easily understood by the public, even if estimates of effects on health outcomes are rough. And we have had guidelines for decades with little to no improvement in population levels of activity ... so how does the authors' work help address this problem?

Response to: NCOMMS-24-69585-T

Title: Wearable Device-Based Health Equivalence of different Physical Activity Intensities against mortality, cardiometabolic disease, and cancer

REVIEWER (#)	General Comment	Author Response & Changes Made
Reviewer 1		
1	Key results This a large retrospective cohort study of 73,485 adults, aged 40 to 79 years investigating the equivalence of wrist-worn wearable-device derived physical activity intensity levels on 6 outcomes of all-cause (ACM) and cardiovascular disease (CVD) mortality; major adverse cardiovascular events (MACE); type 2 diabetes incidence; and cancer mortality and incidence. VPA had a significant dose-response with ACM, CVD mortality, MACE and T2D. MPA had a dose response relationship with outcomes up to 30 minutes, and LPA was significantly associated with T2D and ACM. One-minute of vigorous intensity physical activity was equivalent to about 4-9 minutes of MPA and 53-156 minutes of LPA for all-cause mortality and MACE, type 2 diabetes and CVD mortality. The methods, analyses and results are sound and provide a contribution to the field. However, I have some concerns with the implications of the findings in the Discussion. These findings are of interest on risk mitigation specific to this sample of individuals, but it does not seem these findings can support a large worldwide public health initiative, as suggested, to change PA guidelines in light of these findings.	We would like to thank the reviewer for the positive remarks. We provide full responses to their specific concerns on the translation of our findings below.

2	Abstract: The introduction discusses assessing equivalence of LPA and MPA against VPA, but doesn't discuss equivalence of these metrics on what outcomes. Please add this in. The conclusion states that findings support wearable device data-based guidelines, which feels like a bit of a jump from the results of the study.	We have now clarified that the outcomes of interest are all-cause (ACM) and cardiovascular disease (CVD) mortality; major adverse cardiovascular events (MACE); type 2 diabetes incidence; and cancer mortality and incidence. This text was previously in the Methods section but has been moved to the "Background and Objectives" part of the abstract (page 2). "Using a large wearables dataset and a highly granular (10-sec epoch) machine learning-based intensity classifier, we assessed the equivalence of light intensity (LPA) and MPA against 1 minute of VPA in terms of all-cause (ACM) and cardiovascular disease (CVD) mortality; major adverse cardiovascular events (MACE); type 2 diabetes incidence; and cancer mortality and incidence." We have revised the Abstract Conclusion to emphasise that these findings should be incorporated into the evidence base to better inform future physical activity guidelines rather than entirely wearable device-based guidelines (page 2). "In this work, we found that each minute of vigorous intensity activity is roughly equivalent to 4-9 minutes of moderate and 53-156 minutes of light intensity physical activity, suggesting a considerable departure from previous self-report conventions. These findings highlight the need to incorporate device-based health equivalence estimates into evidence syntheses to inform future physical activity guidelines, as well as algorithms for consumer-grade wearable devices."
3	Introduction: Line 55 Risk reduction for what? Metabolic equivalent needs to have an acronym denoted in the first mention The first paragraph needs to have more mention of the benefits of PA and intensity levels on what particular condition/disease etc. It is too broad to just state "benefits" The lower time benefit of VPA is a noted benefit, however, authors should discuss the higher intensity levels needed to attain VPA, particularly for the "patient" population	Line 56, We have provided clarification plus we have updated the references. We have provided additional clarification and references to support the discussion of "benefits". We have revised the section on both the opportunities and challenges vigorous intensity presents and have added appropriate references. Please refer to the revised section (page 3) as follows: "There is a clear dose-response relationship between physical activity intensity and mortality or non-communicable disease risk reduction." "Studies examining the associations of physical activity volume with mortality and other long term health outcomes have thus assumed that one minute of vigorous activity produces similar benefits to around two minutes of moderate activity."

4	Line 81-83 should mention equivalence of LPA and VPA on what conditions	We have added appropriate clarification including references, please refer to page 3 for the revised as follows: “The few studies that used devices to quantify the association of LPA with mortality or cardiovascular disease show mixed findings, and none estimated specifically its per-time unit equivalence to vigorous intensity.”
5	Line 84 – What are the health values that are assigned? These should be spelled out	We have added the following as clarification (plus references) in page 4 of the revised draft: “Using algorithms that are partly based on existing, less empirical, research standards, consumer grade wearables commonly assign “health value” to each minute of physical activity the wearable device records, to support setting and monitoring physical activity goals, e.g., Google Fit Heart Points and Apple Fitness+. For example, Google Fit assigns Heart Points based on the intensity of the activity: 1 heart point if it is moderate intensity (100-130 steps per minute) and 2 heart points if it is vigorous (>130 steps per minute).”
6	Line 86 – 89 The algorithms in wearables are not calculating PA metrics base don health outcomes data; This doesn’t fully make sense the way it is written. Perhaps if the above comment is elaborated upon, it may make more sense the way it is currently written.	We have revised the section in page 4 of the revised draft: “In the absence of published epidemiological studies estimating the equivalence of wearable device-measured physical activity intensity, it is highly unlikely that any proprietary consumer-grade algorithms reflect the associations of intensity with long term clinical and mortality endpoints when calculating these metrics.”
7	Methods: Non-wear time procedures should be written out instead of just referenced Can the authors list out what cancers are associated with increased PA? And also discuss why these cancers were chosen in lieu of others?	We have added the following (plus appropriate references) to add clarity on non-wear time procedure in page: “The data were calibrated, and non-wear periods were identified according to standard procedures. Non-wear periods were identified using the signal standard deviation to detect periods where the standard deviation was below 0.13 milligravity for 30 or more minutes, reflecting non-wear. Sleep was detected based on relative changes in wrist tilt angle.” Physical activity related cancers include those for which epidemiological evidence indicates an association with increased levels of physical activity. For our study, we focused on those cancer types where leisure-time physical activity has been most consistently evaluated in the literature. This selection was informed by prior meta-analyses and pooled studies that demonstrated significant or suggestive associations between physical activity and reduced risk. Cancers included in our analysis are colon, breast (postmenopausal), endometrial, esophageal adenocarcinoma, liver, kidney, stomach (cardia), bladder, head and neck, and myeloid leukemia, among others. A full list of included cancer types is provided in eTable 2 in the Supplement.

		We excluded in situ cancers, benign tumors, and cancers with uncertain, non-melanoma skin, or non-well-defined classifications because these categories either lack a clear invasive or malignant phenotype, are not systematically reported in most cohort studies, or exhibit biologically distinct pathways less likely to be influenced by physical activity. Non-melanoma skin cancer, for instance, is primarily driven by ultraviolet exposure rather than systemic risk factors like physical activity. Similarly, in situ cancers, while potentially informative for specific analyses, often represent precursors rather than fully developed malignancies, and their inclusion could dilute associations observed with invasive cancers. By selecting cancers with robust prior evidence and excluding less biologically or methodologically relevant categories, we aimed to enhance the precision and generalizability of our findings, ensuring our results have clear implications for public health and cancer prevention strategies. For clarity, the following text has been incorporated into the manuscript on page 6: “Physical activity related cancer encompasses various cancer types associated with increased levels of physical activity, excluding in situ, benign, uncertain, non-melanoma skin cancer, or non-well-defined cancers. The cancer types were selected informed by prior meta-analyses and pooled studies that demonstrated significant or suggestive associations between physical activity and cancer risk, detailed list of cancer ICD-10 codes in eTable 2.”
8	Analyses The rationale for excluding patients with prevalent disease (what diseases were these?) should be discussed. These exclusion criteria may be better suited in a section listing out the patient inclusion/exclusion criteria rather than in the analyses section	We created a new subsection titled “inclusion criteria” in the methods, including references (page 6). Following revision has been added: “Inclusion criteria We followed established inclusion criteria from analogous analyses. For each outcome, participants with a prior diagnosis of the respective outcome at or before the accelerometry baseline were excluded. Adjustments in the models primarily accounted for cardiovascular disease (CVD) and cancer, as these are the leading causes of mortality. Specifically, analyses for all-cause mortality (ACM) and type 2 diabetes were adjusted for prior diagnoses of cancer and CVD. For major adverse cardiovascular events (MACE) and other CVD outcomes, prior cancer incidence was considered, while analyses for cancer outcomes were adjusted for previous CVD diagnoses.”

9	Discussion: Major concern Given the vast majority of adults do not meet the current recommended PA guidelines, it seems counterintuitive to increase the required minutes of MPA, for example, based on these findings, Authors may give some consideration to how these findings could be practically used to support PA promotion, rather than the notion that more is better or a higher intensity level is better. The Discussion could elaborate more on this aspect, and try to better align these findings with public health messaging, or find a way to integrate them into more of a precision medicine approach.	Our intention was not to be evangelical about higher intensities or to promote “the notion that more is better or a higher intensity level is better”. We have revised the opening paragraph of discussion in page 10 to indicate ways how our findings expand behaviour change options, including relevance to precision medicine: “Using a wide array of mortality and major non-communicable disease end points, our study is the first that to systematically quantify the equivalence between LPA, MPA, and VPA various intensities of physical activity for use in public health and prevention research, as well as in consumer-grade wearable devices. Our work expands behaviour change options by indicating alternatives to higher intensities, informs the development of physical activity guidelines based on wearable devices, enhances surveillance measures, and supports practitioners and future trials to establish more accurate dosages and define options for prescribing physical activity prescription and personalised medicine initiatives.”
10	Lines 274-276 It may be a stretch to say this is providing clinicians with information to select interventions, when interventions are not commonly broken down into LPA, MPA, VPA etc. The information given about a possible intervention in lines 271-274 is more helpful, and further examples of who the high-risk patients are and how precision medicine using this approach and these data can be used to mitigate risk for these patients.	We appreciate your point regarding the practical application of our findings in clinical settings. We believe, however, that our assertion about the potential impact of our work is well-supported. Physical activity interventions are indeed often tailored to specific intensity levels, with moderate to vigorous intensity physical activity (MVPA) frequently being a primary focus. For instance, randomised control trials, such as https://www.bmj.com/content/376/bmj-2021-068465, emphasize the role of MVPA in improving health outcomes. We hope this helps clarify the relevance of our approach and findings.
11	Limitations Authors discuss the low response rate and unrepresentativeness to the target population as not affecting the estimate of PA with mortality, but what about the other conditions? And how do these factors weigh into the development of recommendations for wearable devices and adjustments to the existing PA guidelines?	Based on the statistical underpinning of our cited study¹, there is no reason to believe that cohort non-representativeness would affect differentially e.g. incident disease outcomes. To address the concern raised, however, we used more cautious wording and revised in page 13: “However, evidence suggests that, at least with mortality endpoints, low response rates, and the subsequent unrepresentativeness to the target population, does not materially affect estimates of the association of physical activity.” Reference: 1. Stamatakis, E. et al. Is Cohort Representativeness Passé? Poststratified Associations of Lifestyle Risk Factors with Mortality in the UK Biobank. Epidemiology 32, 179–188 (2021).

Reviewer 2		
1	Thank you for the opportunity to review this interesting theoretical study. I did appreciate the authors' attempt to consider a long-standing issue in physical activity epidemiology. However, due to some major methodological limitations, I feel this work has limited public health policy or practice relevance.	We would like to thank the reviewer for taking the time to review our work and give us feedback on how to improve it. We have provided a response to each specific comment below.
2	Major concern 1: the authors correctly assert that the doubling of time allocated for vigorous-intensity physical activity (vs moderate) is a rough estimate at best. However, using data collected by devices does not allow examination of differences in energy expenditure and their relative contribution to chronic disease risk and mortality. Accelerometers measure acceleration (movement in milligravities), which is an objective measure of movement. However, without other measures to give an indication of energy being expended (even a proxy like heart rate monitoring) it is impossible to estimate energy expenditure. A 40 year old male with good cardiorespiratory fitness might experience movement captured within the algorithm for 'moderate' physical activity as 'light' intensity (based on energy expended). Whereas an obese 60 year old male recording the same acceleration (which the algorithm defines as 'moderate') might actually be expending significantly higher energy. The authors may address some of this confounding through the adjustment of their models, but BMI or another measure of body composition	Major concern 1 Here the reviewer is querying the approach used by the field of physical activity epidemiology in its entirety. For example, the vast majority (if not all) of the evidence used for guidelines development uses absolute intensity standards, see for example WHO 2020¹ and Physical Activity Guidelines for Americans reports². The use of wearables featuring heart rate sensors in epidemiology are a rare exception, and to our knowledge, no major population cohort has captured such data at scale. As the reviewer rightly implies, physical activity epidemiology primarily relies on absolute intensity thresholds derived from accelerometer data. The UK Biobank represents the largest and most comprehensive resource of this kind currently available. We agree that, in an ideal world, the use of relative intensity would be optimal, but this would require concurrent assessments of cardiorespiratory fitness at around the same time the wearable devices were worn. Unfortunately, such data are not available within the UK Biobank. In any case, previous empirical evidence by our team suggests that the concerns of the reviewer may have little practical impact on our findings, and by extension on the equivalence estimates. We have shown that the use of absolute or corrected (for BMI, age, sex) intensity does not materially influence the association of physical activity volume with mortality³. In our current analyses, it would have been inappropriate to adjust for BMI due to its role as a potential mediator on the casual pathway between physical activity and long-term endpoints (particularly for LPA and MPA). In previous UK Biobank analyses of ours we have

	was not included. I understand that BMI can be considered a mediator between physical activity and chronic disease, but we are still left with a very rough approximation of energy expenditure using the device-based measures. Being able to report acceleration (milligravities) comparisons for different disease endpoints is not amenable to informing public health practice. Sometimes very rough – but easily understood – concepts are more useful for health promotion.	carried out analyses with and without BMI and we could see no material difference in findings⁴. We have added the above as a limitation, and cited the above reference: “The wearables used in the UK Biobank relied on accelerometers quantifying absolute intensity, not energy expenditure. Previous research, however, has indicated that adjusting absolute intensity for participant characteristics that determine energy expenditure (weight, height, age, and sex), does not materially influence the association of physical activity volume with mortality.” Reference:  1. Piercy KL, Troiano RP, Ballard RM, Carlson SA, Fulton JE, Galuska DA, George SM, Olson RD. The Physical Activity Guidelines for Americans. JAMA. 2018 Nov 20;320(19):2020-2028. doi: 10.1001/jama.2018.14854. PMID: 30418471; 2. Bull, F. C. et al. World Health Organization 2020 guidelines on physical activity and sedentary behaviour. Br J Sports Med 54, 1451–1462 (2020). 3. Martenstyn, J. A., Powell, L., Nassar, N., Hamer, M. & Stamatakis, E. Intensity-Weighted Physical Activity Volume and Risk of All-Cause and Cardiovascular Mortality: Does the Use of Absolute or Corrected Intensity Matter? J Phys Act Health 16, 1054–1059 (2019). 4. Stamatakis, E. et al. Vigorous Intermittent Lifestyle Physical Activity and Cancer Incidence Among Nonexercising Adults: The UK Biobank Accelerometry Study. JAMA Oncol 9, 1255–1259 (2023).
3	Major concern 2: by excluding participants with prevalent chronic diseases of interest (e.g. physical activity related cancers) from the study population, the authors have attempted to answer a causal question about (mortality) outcomes that are defined conditional on post-exposure events. When estimating effects of physical activity intensities on cancer-related mortality, participants are only eligible for inclusion if they survive to the development of cancer (which must occur before cancer-related death). Depending on when follow-up commences this can introduce selection bias	As is standard in these analyses, follow up commences upon completion of the wearables measurement. To address concerns regarding potential selection bias and/or immortal time bias in our analysis, we conducted two additional analyses to evaluate the robustness of our findings. The results of these analyses are presented in Response Figures 1 and 2 below. Adjustment for Previous Events: In the first analysis, we compared the dose-response relationship from the original model to a new model where participants with prior events were not excluded but instead a cancer diagnosis was adjusted for as a covariate. The overlapping dose-response curves (Response Figure 1) demonstrate consistent results between the two approaches, supporting the validity of our original findings.

and/or immortal time bias. These biases can change effect estimates considerably.

Response Figure 1: Adjusted dose response associations of daily incidental vigorous (VPA), moderate (MPA), and light (LPA) intensity physical activity with physical activity related cancer mortality and physical activity related cancer incidence. The solid line represents the original model, while the dashed line depicts the revised model adjusted for previous cancer events.

Panel A: physical activity related cancer mortality (original model: n = 67,499; events = 538; adjustment for previous events: n = 73,485; events = 704), Panel B: physical activity related cancer incidence (original model: n = 68,828; events = 2,662; adjustment for previous events: n = 73,031; events = 3,135).

Extended Exclusion Period for Initial Events:

In the second analysis, we extended the exclusion period for disease-specific analyses. Specifically, we compared the dose-response relationship from the original model (excluding individuals registering an event within the initial 12 months of follow-up) with a new model where individuals registering an event within the first 36 months of follow-up were excluded. The overlapping dose-response curves (**Response Figure 2**) again show a high degree of concordance, indicating that our results are robust to variations in the follow-up exclusion criteria.

Response Figure 2: Adjusted dose response associations of daily incidental vigorous (VPA), moderate (MPA), and light (LPA) intensity physical activity with physical activity related cancer mortality and physical activity related cancer incidence. The solid line represents the original model, while the dashed line depicts the revised model refers to exclusion of first 36 months of follow-up events.

		Panel A: physical activity related cancer mortality (original model with exclusion of 12 months of events: n = 67,499; events = 538; extended exclusion period: n = 68,720; events = 452), Panel B: physical activity related cancer incidence (original model with exclusion of 12 months of events: n = 68,828; events = 2,662; extended exclusion period: n = 67,841; events = 1,957).
4	Minor comment: the authors explain in the methods that they excluded participants with prevalent chronic diseases of interest. However, some models adjust for a cancer diagnosis, or diabetes. Does this mean that the adjustment is based on incident disease that occurred after the accelerometer baseline assessment? Please clarify this point.	We created a new subsection titled “inclusion criteria” in the methods, including appropriate references (page 6), to clarify this. Following revision has been added: “Inclusion criteria We followed established inclusion criteria from analogous analyses. For each outcome, participants with a prior diagnosis of the respective condition by the time the accelerometry baseline started were excluded. Adjustments in the models primarily accounted for cardiovascular disease (CVD) and cancer, as these are the leading causes of mortality. Specifically, analyses for all-cause mortality (ACM) and type 2 diabetes were adjusted for prior diagnoses of cancer and CVD. For major adverse cardiovascular events (MACE) and other CVD outcomes, prior cancer incidence was considered, while analyses for cancer outcomes were adjusted for previous CVD diagnoses.”

Comments	Responses
Reviewer 2	
R2.1 I thank the authors for the time they have taken to respond to my comments. However, I think they have misunderstood my first point: even though accelerometry is an improvement upon self-reported data in many ways, the assignment to categories of 'light', 'moderate' and 'vigorous' is still based on assumptions that may not be correct at the individual level. But without collapsing the data in such a way accelerometer data are meaningless to consumers I simply wanted the authors to explain more thoroughly why the categories of accelerometer-measured activity, and the relative volumes compared to vigorous activity, will improve public health policy and programmes.	We apologise to the reviewer for our misinterpretation of their comment and thank them for providing clarification. We compared intensities and volumes relative to vigorous activity to align with prevailing public health messaging—the 2:1 ratio of moderate to vigorous intensity—that has informed guidelines, policies, and programs since its introduction in the 2008 US physical activity guidelines. A frequently overlooked limitation of this prevailing ratio is that it was based largely on anecdotal evidence rather than on empirically derived data. An important aspect of our study that positions it to influence policy and programs is its alignment with recent shifts in the priorities of guideline bodies. A new WHO report (1) indicates the plans to move toward incorporating more weight for wearable-derived evidence into the upcoming 2030 guidelines. Notably, the report also highlights a necessity for the research field to revisit the health equivalence of vigorous intensity to moderate intensity in addition to light intensity - where evidence is more scarce – with disease risk using wearables-based evidence. We have included the potential impact of our study to improve public health policy and programmes in the Discussion Section on lines 268 – 280. To fully address the reviewer’s comment, we have updated these lines to now explicitly refer to policy and programs: “Our equivalence findings between vigorous and moderate intensity are broadly consistent with randomised controlled trials and cross-sectional studies. Previous works

	that have compared vigorous intensity to moderate intensity activities and exercise under supervised and controlled clinic conditions³⁶⁻⁴⁰ demonstrated an approximate 1:7 to 1:13 ratio for surrogate cardiometabolic outcomes such as lipid profile, blood pressure, and cardiorespiratory fitness⁴²⁻⁴⁶. Our results expand this evidence with long term clinical endpoints and mortality in a real-world population cohort measured in environments outside controlled clinic conditions. Our findings at the population level alongside those of smaller RCTs under controlled conditions provides evidence that may improve future public health messaging and programs. Taken together, our findings prompt some reinterpretation of the physical activity evidence-base, particularly in regard to current guidelines suggesting 75-150 minutes of vigorous intensity is equivalent to 150-300 minutes of moderate intensity, reflecting the traditional 1:2 ratio between vigorous and moderate intensity.” 1) World Health Organization. Physical activity measurement and surveillance in adults: report of a scoping and planning meeting, 27-28 November 2023. Geneva: World Health Organization 2024:iv, 34 p.
R2.2 The guidelines based on self-reported estimates are easily understood by the public, even if estimates of effects on health outcomes are rough. And we have had guidelines for decades with little to no improvement in population levels of activity so how does the authors' work help address this problem?	In the current public health guidelines (which are largely based on self-report estimates), messaging is communicated using minute-based metrics which are easily understood by the general public. Our study leverages wearable-derived data—also expressed in minutes of activity—to maintain continuity with existing messaging conventions, while adding critical nuance and a more evidence-based approach.

We differentiate the health impacts of vigorous, moderate, and light intensity activities, moving beyond the current simplistic messaging of “more is better” mantra and the potentially daunting, for many inactive adults at least, threshold of 150 minutes of MVPA per week.

By showing a spectrum of activity patterns that reduce risks of mortality, type 2 diabetes, CVD, and cancer, our findings can be used to empower individuals to adopt flexible strategies that fit within their capacity and preferences. This approach may potentially promote more sustainable activity patterns in the long term, ultimately leading to improved adoption, adherence, and hopefully increases in overall physical activity.